# Genotype by Environment Interaction in Grain Iron and Zinc Concentration and Yield of Maize Hybrids under Low Nitrogen and Optimal Conditions

**DOI:** 10.3390/plants12071463

**Published:** 2023-03-27

**Authors:** Sajjad Akhtar, Tesfaye Walle Mekonnen, Gernot Osthoff, Kingstone Mashingaidz, Maryke Labuschagne

**Affiliations:** 1Department of Plant Sciences, University of the Free State, Bloemfontein 9300, South Africa; 2Department of Microbial Biochemical and Food Biotechnology, University of the Free State, Bloemfontein 9300, South Africa; 3ARC-Grain Crops, Potchefstroom 2520, South Africa

**Keywords:** biofortification, iron, GEI, low N, maize, zinc

## Abstract

Maize is the staple food crop for millions of people in sub-Saharan Africa. Iron (Fe) and zinc (Zn) deficiency is a significant health risk that mainly affects low-income populations who rely solely on maize-based diets. This problem can be alleviated by developing micronutrient-rich maize grain. The aim of this study was to determine the adaptation and performance of hybrids for Fe and Zn concentration and grain yield under low soil nitrogen (N) and optimal conditions. Eighteen hybrids derived from lines and testers with low, medium, and high Fe and Zn concentration were grown during the summer rainy seasons of 2017 and 2018 at three locations under low and optimal N conditions. There were significant genotype and environment effects for grain yield, and Fe and Zn concentration, but the genotype by environment interaction effects were the largest, accounting for between 36% and 56% of variation under low N conditions. Low N levels significantly reduced grain yield, and Fe and Zn concentration. Hybrids G1, G2, G4, G7, G10, G11, and G16 were relatively stable, with relatively high mean Fe and Zn concentrations, and low additive main effects and multiplicative interaction (AMMI) stability values and iron stability index (FSI) and zinc stability index (ZSI) under low N conditions. These genotypes can be considered for production under low N stress conditions. Two environments (E4 and E3) were identified for good discriminatory power for genotype performance in terms of Fe and Zn content, respectively. Stable and high-yielding genotypes with high Fe and Zn concentration can be used as biofortified hybrids, which can contribute to a sustainable solution to malnutrition in the region, especially under low N conditions.

## 1. Introduction

Maize (*Zea mays* L.) is the third most important cereal crop in the world [1,2] and the most important food security crop for low-income countries in sub-Saharan Africa (SSA) [2]. It is the primary source of calories for humans in this region and contains micro-nutrients, lipids, minerals, and vitamins [3,4]. The typical maize kernel, on dry weight basis, is composed of 61–78% starch, 6–12% protein, 3.1–5.7% oil, 1.0–3.0% soluble sugars, 1.1–3.9% ash, 1.5–2.1% crude fiber, 4.3–4.5% fat, and its energy value is 1527.16 KJ 100 g^−1^ [4,5].

Micronutrient malnutrition is affecting billions of people worldwide [6]. Among the micronutrient deficiencies, Fe and Zn deficiency are the most common in the developing world [7]. Although maize is the most important source of calories in SSA, deficiencies of Fe and Zn in maize-based diets pose serious health challenges in developing countries [8,9,10,11]. Fe deficiency is the predominant cause of anemia, affecting 27% of the global population [12]. Fe deficiency causes anemia, low cognitive functioning, immune suppression, fatigue, low birth weight of infants, and increased mortality and morbidity [2,13]. Zn deficiency causes delayed and reduced growth, diarrhea, skin inflammation, acute respiratory infection, hypogonadism, epidermal disorders, and dysfunction of the immune and central nervous system [2,14]. This deficiency largely affects women, and children below five, who suffer from severe acute malnutrition which is quite common [2,15,16].

In response to these challenges, genetic biofortification of maize is a feasible, effective, safe, and sustainable micronutrient delivery approach compared to agronomic fortification, postharvest food fortification, a diversified diet, and oral diet supplementation [10]. It is a promising, cost-effective, and sustainable approach for delivering micronutrients to populations with limited access to diverse diets and other micronutrient interventions [17]. Biofortification, through conventional and molecular-based breeding techniques, can increase the concentration and bioavailability of Fe and Zn in the grain [10,17]. In South Africa, maize production is practiced under diverse environmental conditions of varying soil structure, soil fertility, temperatures, disease pressure, altitude, and longitude, planting time, and rainfall distribution [18,19].

Low N levels are one of the major factors for yield reduction in maize fields in SSA [20]. Nitrogen is an essential macro element, which plays a crucial role in determining the yield of maize in the tropics and elsewhere [21]. Most small-scale farmers in SSA grow their maize under low N conditions due to a lack of fertilizer, and the effect of this on the Fe and Zn levels in maize grain is largely unknown. Protein in grain constituents of N is influenced by the soil N conditions. A lack of dietary protein in general adversely affects infants’ overall well-being, and symptoms include diarrhea and severe wasting, collectively known as “kwashiorkor” [22]. Any biofortification efforts should take this reality into account. Therefore, an understanding of the genetic diversity among maize genotypes for low N tolerance and Fe and Zn variation under low N conditions offers an opportunity to develop maize hybrids that possess tolerance genes to low N, which is critical for sustainable maize production in areas with low soil fertility [20,23], and to select for high Zn and Fe levels under low N conditions. Many methods have, in the past, been used for stability and adaptability analyses of genotypes in multi-environment trials (MET) [24,25,26]. Some commonly used methods include genotype main effect plus genotype by environment interaction (GGE) biplots [27], the additive main effects and multiplicative interaction (AMMI) analysis [28], AMMI stability value [29], and yield stability index used for quantifying and measuring the magnitude of genotype environment interaction (GEI) [30].

The aim of this study was to determine the expression of, and genotype by, environment interaction effects of Fe and Zn concentration and yield of 18 maize hybrids developed from lines and testers with low, intermediate, and high Fe and Zn concentration, under low and optimal N conditions.

## 2. Results

### 2.1. Genotype and Environmental Variance

The analysis of variance showed highly significant effects (*p* < 0.001) of genotype, environment, and environment by genotype interaction on grain yield, Fe, and Zn under both N conditions across environments (Appendix A).

### 2.2. AMMI Analysis

The AMMI analysis (Table 1 and Table 2) showed that genotype, environment, and GEI effects were highly significant (*p* < 0.001) under both N conditions for grain yield, Fe and Zn. Under low N conditions the environment accounted for 19.77%, while genotype and GEI accounted for 24.58% and 49.71% of the variation observed in grain yield. The relative contribution of genotype, environment, and GEI variances to the total sum of squares was 21.87%, 34.01%, and 42.71% for Fe concentration in grain and 38.85%, 12.11%, and 36.54% for Zn concentration in grain. The two IPCAs combined accounted for a total of 90.81%, 89.12%, and 90.87% of the observed variation due to GEI for grain yield, Fe and Zn, respectively.

Under optimum conditions, genotype, environment, and GEI accounted for 17.87%, 66.95%, and 12.37% of total variation for grain yield, 20.31%, 21.45%, and 55.92% for Fe and 22.03%, 43.92%, and 30.46% for Zn. The two IPCAs accounted for a total of 89.11%, 84.26%, and 92.65% of the observed variation due to GEI for grain yield of Fe and Zn, respectively.

### 2.3. Delineation of Mega Environment and Superior Genotype for Grain Yield, Iron, and Zinc

#### 2.3.1. Which-Won-Where and What

The polygon view of the genotypes in the GGE biplot for 18 hybrids are presented in Figure 1A–C (low N conditions) and Figure 2A–C (optimum conditions). Under low N conditions, for grain yield, there were two mega environments. The first mega environment comprised E1 and G1, G7, G13, and G16, and had the most adapted and highest yielding genotypes (Figure 1A). The genotypes G2, G6, G8, G10, G11, G12, and G14 were specifically adapted to mega environment two (E2 and E3).

For Fe content (Figure 1B), E2, E3, and E4 were clustered in the first mega environment, while the other environments comprised mega environment three. G4, G12, and G15 were the best performing and the most adapted genotypes for mega environment one. Genotypes G1, G2, G3, G7, and G10 were adapted to mega environment two (Figure 1B). For Zn, E2, E3, and E4 were clustered in the first mega environment (Figure 1C).

Under optimum N conditions, genotypes G4, G7, G11, G12, and G15 were the best performing and adapted to all environments for grain yield (Figure 2A). E1 and E4 environments were presented in one mega environment, and it comprised only four genotypes for Fe content (Figure 2B). For Zn, the first mega environment comprised E1, E2, and E3; genotypes G4, G7, and G10 were adapted to it, while G6, G8, and G18 were winner genotypes in E4 (Figure 2C).

#### 2.3.2. Ideal Genotypes

The GGE biplot (Figure 3A) identified G1 and G16 as ideal high-yielding and stable hybrids across the environments because they fell close to the center, as ideal hybrids under low N conditions. Based on the high Fe concentration in grain and stability performance, G4 was the best-performing and stable genotype under the same conditions (Figure 3B). Moreover, G11 was an ideal genotype with high Zn concentration in grain, which fell close to the center of the concentric circle (Figure 3C). Based on the average environment coordination (AEC) method, G18 for grain yield, G8 for Fe, and G6 for Zn were the most unstable across the environments under both N conditions (Figure 2A–C).

Under optimum N conditions, the GGE biplot (Figure 4A–C) identified G4 and G7 for grain yield, G14 for Fe, and G10 for Zn as ideal high yielding and stable genotypes across the environments because they fell close to the center as superior genotypes under optimum conditions. On the other hand, G18 for grain yield, G7 for Fe, and G8 for Zn were identified as unstable genotypes across environments under the optimum conditions.

#### 2.3.3. Ideal Environments

Environments, E4, E3, and E1 for Fe, Zn, and grain yield, respectively, were close to the concentric circle. They were the most ideal and powerful to discriminate between the performance of the genotypes (Figure 5A–C) under low N conditions across environments. Under optimum N conditions, environments E1 for grain yield, E4 for Fe, and E3 for Zn were closest to the epicenter, which represents the ideal environments, and the ideal environments offer the highest discriminatory power (Figure 6A–C).

### 2.4. Stability Analysis Using IPCA, AMMI Stability Value (ASV), and Yield Stability Index (YSI), Iron Stability Index (FSI), and Zinc Stability Index (ZSI)

Under low N conditions, the genotypes with positive and high values of IPCA are considered as stable, while negative values of IPCA are considered as unstable. Accordingly, genotypes G10, G11, and G2 for grain yield, G13, G18, and G9 for Fe and G4, G13, and G2 for Zn had the most positive IPCA1 scores in the studied environments under low N conditions (Table 3). However, genotypes G8, G7, and G1 for grain yield, G1, G17, and G7 for Fe and G18, G3, and G14 for Zn exhibited large and negative IPCA1 scores with studied environments under low N conditions.

ASV, ASVF, and ASVZ of the hybrids across the environments varied from 0.26 (G13) to 3.08 (G18) for grain yield, 0.35 (G5) to 3.83 (G1) for Fe, and 0.19 (G16) to 2.29 (G4) for Zn.

Based on this model, genotypes with the lowest values of ASV, ASVF, ASVZ, YSI, FSI, and ZSI or that have the smallest distances from the origin are considered as the most stable, whereas the highest values of ASV, YSI, FSI, and ZSI are considered as unstable.

Based on ASV, the most stable (low ASV and GEI) genotypes were G13, G3, G14, G18, and G12, while genotypes such as G10, G8, G11, G17, and G2 were the least stable for grain yield. According to the results of ASVF, genotypes G5, G11, G14, G12, and G16 were the most stable genotypes for Fe across the environments (low ASVF and GEI). However, G1, G17, G13, G7, and G18 were the most unstable genotypes across environments. ASVZ results indicated that genotypes G16, G17, G10, G8, and G6 were the most stable genotypes for Zn across the environments, while genotypes such as G4, G18, G13, G3, and G14 were the least stable (Table 3).

The lowest YSI is considered the most stable with high mean values. Accordingly, G13, G16, G6, G3, and for grain yield, G12, G11, G2, G15, and G4 for Fe and G10, G11, G16, G1, and G15 for Zn were the most stable genotypes with high mean grain yield, Fe, and Zn under low N conditions (Table 3).

Under N conditions for IPCA1 scores, genotypes G10, G18, and G2 for grain yield, G7, G8, and G3 for Fe, G10, G18, and G2 for Zn had the largest positive interaction with the studied environments. Hybrids G17, G4, and G11 for grain yield, G15, G14, and G10 for Fe and G17, G4, and G11 for Zn displayed large and negative interactions with the studied environments.

ASV, ASVF, and ASVZ of the genotypes across the environments ranged from 0.27 (G3) to 5.24 (G4) for grain yield, 0.06 (G12) to 5.11 (G7) for Fe, and 0.68 (G3) to 3.85 (G6) for Zn (Table 4). From ASV, G3, G16, G6, G1, and G14; from ASVF, G12, G4, G17, G13, and G18; and from ASVZ, G3, G14, G13, G12, and G16 were the most stable across environments. However, G4, G10, G11, G17, and G18 for grain yield, G3, G6, G7, G14, and G15 for Fe and G5, G6, G 8, G9, and G18 for Zn were the most unstable genotypes (Table 4).

Based on the YSI, FSI, and ZSI values, genotypes G11, G17, and G18 were the most unstable and had low mean values across the test environments for grain yield, Fe, and Zn, respectively. G6, G13, and G3 were the most stable hybrids for grain yield Fe and Zn, respectively, under optimum N conditions across environments. G18, G7, and G8 were the least stable genotypes across the environments for grain yield Fe and Zn (Table 4).

## 3. Discussion

Breeding for combined high Fe and Zn concentration and grain yield under low nitrogen and optimal conditions is challenging due to the polygenic inheritance and environmental influence on these traits. The selection of stable genotypes through evaluation in diverse environments under targeted stress conditions is important for varietal development. Genotype, environment, and their interaction reduced the Fe and Zn concentration in grain, as was also previously reported in pear millet [26]. The current study observed wide genetic variation among the evaluated genotypes for Fe and Zn concentration under both optimal and low N conditions, indicating the potential for selection for high levels of these minerals. Grain yield variation under low and optimum N conditions across the environments also demonstrated the opportunity for the selection of higher yielding genotypes with increased Fe and Zn concentration in grain and, in addition, to develop low N tolerant maize materials.

The GEI effect accounted for between 36% and 50% of the total variation of grain yield, and Fe and Zn under low N conditions, while the genotype and environmental effects accounted for less than 25% of the variation for grain yield. The GEI effects accounted for more than 54% of the variation for Zn content under optimum conditions. GEI was reported to influence micronutrient concentrations, affecting their uptake by roots, translocation through shoots, and assimilation in grain [31]. A better understanding of the GEI will improve the selection process for the targeted environments. These results concur with a study [32], which reported that grain yield stability was affected by divergent environmental conditions, which resulted in high GEI. Contrary to this, another study [33] reported that the effect of GEI for grain Fe and Zn was small. Varying genotype responses for Fe and Zn concentration in maize grain in different environments were reported in other studies [34,35].

Under low N conditions, the polygon view of the “which-won-where and what or which-is-best-at-what’’ GGE biplot showed that genotypes G4, G7, G8, G10, and G18 for grain yield, G1, G2, G6, and G18 for Fe and G4, G8, G10, G14, and G18 for Zn were the genotypes located furthest from the origin. This indicated that these genotypes performed well under low N stress in specific environments. The other genotypes close to the origin were broadly adapted to all tested environments. The response of genotypes to low N environments for the concentration of Fe and Zn in grain and grain yield were highly variable. Previous studies [33,36,37] also identified mega environments, and found that the concentration of Fe and Zn in grain differs in each genotype across variable environments.

Under optimum conditions, genotypes G4, G10, G17, and G18 for grain yield, G7, G8, G15, and G16 for Fe, and G6, G8, G10, G11, and G14 for Zn were located furthest from the center of origin, showing adaptation to specific environments; meanwhile, the rest of the genotypes showed broad adaptation to the tested environments.

The GGE biplot also compares the genotypes with the ideal genotype located at the epicenter of the concentric circles [38]. Apart from stability, the ideal genotype should have high mean performance [39]. Under low N conditions, the GGE biplot revealed the following ideal genotypes based on high values and stable performance: G1 and G16 for grain yield, G4 for Fe, and G11 for Zn, across the environments. An ideal genotype should have both the highest mean performance and the lowest interactions with the environment (tolerant to low N soil fertility); therefore, these genotypes were located at the epicenter and might be considered desirable genotypes with high mean performance and zero GEI. Because of their genetic background and level of expression of measured traits, ideal genotypes do not always exist in reality. The N effects were significant on the performance of genotypes across the environment for grain Fe and Zn concentration. In general, the concentration of Fe and Zn in grains under optimum conditions was higher compared to low N conditions. Similar results on Zn concentration in maize grain were previously reported [25].

The ideal environment should be located at the epicenter of the circles of the GGE biplot. Under low N conditions, E4 (for grain yield), E3 (for Fe), and E1 (for Zn) were identified as the ideal environments. Identifying the best test environment is essential for selecting potential maize hybrids. The ideal environments have better discriminative power and are representative across the target environments. Hence, these environments allow the genotypes to express their genetic potential under low N stress conditions, which enables efficient selection. In addition, these environments must be considered a testing environment for low N stress tolerance development for grain yield and concentration of micronutrients in grain. Previous studies also identified ideal test environments with good discriminating ability which are representative of the test environment [37,39].

Genotypes with low ASV, ASVF, and ASVZ scores are the most stable; accordingly, G13 for grain yield, G5 for Fe, and G16 for Zn were the most stable genotypes under low N conditions. However, G18, G1, and G4 were unstable for grain yield, Fe and Zn, respectively. Under the optimum conditions, G3, G12, and G3 were the most stable genotypes for grain yield, Fe and Zn, respectively, while G4, G7, and G6 were the most unstable. This indicates that the N levels in the soil significantly influenced the stability of the genotypes for grain yield performance, and Fe and Zn concentration in grain. The GEI magnitude was very different for the two N conditions. The genotypes with the lowest YSI, FSI, and ZSI values are widely adapted and have high values for measured traits [32,40]. Accordingly, G13, G12, and G10 were the most stable genotypes with high mean grain yield, Fe and Zn under low N conditions; meanwhile, G11, G17, and G18 were the most unstable and low yielding. Under optimum conditions, G6, G13, and G3 were the most stable genotypes for grain yield Fe and Zn, respectively, whereas G18, G7, and G8 were the least stable. The crossover stability, grain yield, and concentration of Fe and Zn in grain showed different trends for all genotypes for the two N levels. YSI is recommended for multi-environment trials to identify high-yielding stable genotypes [32].

## 4. Materials and Methods

### 4.1. Descriptions of the Study Area

Potchefstroom is in the northwest province and lies at −26.73° latitude, 27.08° longitude, and at an altitude of 1349 m above sea level (masl), with brown sandy loam soils (Appendix A). Low N conditions were created by depleting soil of N, by planting maize for several years without N fertilization and removing all stover from the field. The fertilizer regime for optimal conditions was compound fertilizer 3:2:1 (25) + Zn applied as a basal application, planting at a rate of 200 kg NPK ha^−1^ to optimum N plots. Limestone ammonium nitrate (LAN) with 28% N was used for top-dressing in two equal splits at 28 and 56 days after emergence at a rate of 100 kg ha^−1^ each, only in optimum N plots. In low N plots, NPK was applied at a rate of 100 kg ha^−1^ once at planting. 

Cedara is in the KwaZulu-Natal province and lies at −29.54° latitude, 30.26° longitude, at an altitude of 1066 masl, with reddish brown clay soils (Appendix A). Fertilizer used was monoammonium phosphate (MAP), 250 kg ha^−1^ at planting, for optimum N environments, and 30 kg ha^−1^ in the low N environment and LAN given at 150 kg ha^−1^ in two equal splits of 75 kg ha^−1^ for only the optimum N sites at 28 and 56 days after emergence. Vaalharts is in the Northern Cape province at −28°06′56.84″ S 24°55′32.50″ E at an altitude of 1192 masl (Appendix A). The fertilizer was applied at the same rate as at Potchefstroom. All standard agronomic practices were applied under both growing conditions. Trials were grown under dryland conditions, which is the norm for the trial areas.

### 4.2. Plant Materials

The planting materials consisted of eighteen maize hybrids which were developed from a line x tester design where three testers (one each with low, intermediate and high Fe and Zn content) were crossed with six lines (two each with low, intermediate and high Fe and Zn content) (Table 5). The parental genotypes were selected after screening 215 South African maize inbred lines obtained from the Agricultural Research Council-Grain Crops (ARC-GC) for concentration (low, intermediate, and high) of Fe and Zn.

### 4.3. Experimental Design and Procedures

The experiment was laid out using a 3 × 6 alpha lattice design with two replications and three incomplete blocks per replication. The distances within rows and between rows were 0.25 m and 0.75 m, respectively, at all locations. In each plot, there were two rows of 4 m length. The plot size was 6 m^2^. All standard agronomic practices (planting depth, weeding, seed rate, and spacing) were applied under both growing conditions. Trials were grown under dryland conditions, which is the norm for the trial areas. Five healthy plants from the middle of each plot were selected for data collection. Soil analysis data are presented in Table 6.

### 4.4. Data Collection

Five plants per plot for all plots were self-pollinated at all locations to generate seed for laboratory analysis in order to eliminate the possibility of pollination with foreign pollen, which may influence results. These samples were oven dried and milled using an IKA, A10 Yellow line grinder (Merck Chemicals Pty Ltd., Darmstadt, Germany) and sieved with a 1 mm screen mesh. The extraction steps of Fe and Zn were performed according to the dry-ashing method outlined by the Association of Official Analytical Chemists (AOAC) [41]. Flour (2 g) was weighed into glazed, high-form porcelain crucibles and ashed in a furnace at 550 °C for 3 h. Then, 1 mL nitric acid (HNO_3_, 55%) was added to the samples for digestion. The samples were then placed in a hot sand-bath until they were completely dry, after which they were returned to the oven for 1 h at 550 °C for further ashing. After cooling, 10 mL of 1:2 HNO_3_ was added to the samples for further digestion. The samples were returned to the hot sand-bath until they became warm. The samples were then transferred to 100 mL volumetric flasks using Whatman # 4 filter paper and filled to the mark with distilled water. Mineral concentrations were measured in triplicate using an Atomic Absorption Spectrophotometer (Agilent Technologies 200 Series AA, New Castle, DE, USA).

Grain yield data were obtained from the central two rows. The grain yield (kg/ha) for every hybrid from the fresh weight data per plot (adjusted to 12.5% moisture) was calculated using the following formula:$$\mathrm{Grain}\;\mathrm{yield}\;\left(\frac{\mathrm{kg}}{\mathrm{ha}}\right)=\frac{\mathrm{Fresh}\;\mathrm{ear}\;\mathrm{weight}\;\left(\mathrm{kg}/\mathrm{plot}\right)\times \left(100-\mathrm{MC}\right)\times 0.8\times 10,100}{\left(100-12.5\right)\times \;\mathrm{area}\;\mathrm{harvested}/\mathrm{plot}}$$

### 4.5. Statistical Analysis

Analysis of variance (ANOVA) across environments was performed [42] using Statistical Analysis Software version 9.4 (SAS) [43]. GEA-R was used for AMMI and GGE biplot [44]. Due to extreme heat and drought conditions in the two seasons, four of the twelve trials (two in each season: Vaalharts optimal in both seasons, and Vaalharts low N in 2018 and Potchefstroom low N in 2017) were lost due to abiotic stress and insect damage. This caused the trials to be unbalanced per year. The trials included in the analysis therefore consisted of four trial by treatment combinations for optimal conditions (Cedara and Potchefstroom, 2017 and Cedara and Potchefstroom, 2018), and four for low N conditions (Cedara and Vaalharts, 2017, and Cedara and Potchefstroom, 2018). For this reason, season was not included as a factor in analysis, but there were rather four trial by treatment combinations under optimal conditions, and four under low N conditions.

The additive main effect and multiplicative interaction (AMMI) model analysis [45] was used for analyzing GEI. AMMI partitions the sum of squares into interaction principal component (IPC) axes. The AMMI analysis of variance summarizes most of the magnitude of GEI into one or few interactions principal component axes (IPCA). The AMMI model equation is given as
(1)$$Y\mathrm{ij}=\mu +G_{i}+E_{j}+\left(\sum K_{n}V_{\mathrm{ni}}S_{\mathrm{ni}}\right)+Q_{\mathrm{ij}}+e_{\mathrm{ij}}$$
where *Y*ij = the observed yield of genotype i in environment j, µ = the grand mean, Gi = the additive effect of the ith genotype (genotype means minus the grand mean), Ej = the additive effect of the jth environment (environment mean deviation), Kn = the eigenvalue of the interaction principal component (IPCA) axis n, Vni and Sni = scores for the genotype i and environment j for the PCA axis n, Qij = the residual for the first n multiplicative com-ponents, eij = the error.

GGE biplots were constructed from the data [45,46,47]. The GGE biplot has many visual interpretations that AMMI does not have, such as visualization of crossover GxE interaction [47]. Moreover, the GGE biplot is more logical for biological objectives in terms of ex-plaining the first principal component score, which represents genotypic level rather than additive level [48]. The GGE biplot is built on the first two major components of a principal component analysis (PCA) using the Site Regression (SREG) model. When the first component is highly correlated with the genotype main effect, the proportion of the yield is considered to be due only to the characteristics of the genotype. The second component represents the variation in the yield due to the GEI [49].

AMMI stability value (ASV) [28] is useful to quantify and rank genotypes according to their yield stability [30].
(2)$$\mathrm{ASV}=\sqrt{\left[{\left[\frac{\mathrm{IPCA}\;1\;\mathrm{Sum}\;\mathrm{of}\;\mathrm{squares}}{\mathrm{IPCA}\;2\;\mathrm{Sum}\;\mathrm{of}\;\mathrm{squares}\;}\left(\mathrm{IPCA}\;1\;\mathrm{scores}\right)\right]}^{2}+{\left[\mathrm{IPCA}\;2\;\mathrm{scores}\right]}^{2}\right]}$$
where ASV = AMMI’s stability value, SS = sum of squares, IPCA1 = interaction of first principal component, IPCA2 = interaction of second principal component.

Yield stability index (YSI) incorporates both mean yield and stability in a single criterion. Low values of both parameters show desirable genotypes with high mean yield and stability [50,51]. The yield stability index was calculated using the following formula; YSI = RASV + R, where RASV is the ranking of the AMMI stability value and R the ranking of genotypes in all environments.

## 5. Conclusions

The performance of the tested genotypes for grain yield, Fe, and Zn concentration in grain was significantly affected by genotype, environment, and their interaction. Genotypes G1, G3, G6, G13, and G16 for grain yield, G2, G4, G11, G12, and G15 for Fe and G1, G10, G11, G15, and G16 for Zn were the most stable genotypes under low N stress conditions. Genotypes G13, G17, and G18 were the stable genotypes for Fe and Zn, and were recommended for optimum N conditions. The AMMI analysis showed that the GEI effect accounted for 49.71% of grain yield, 42.71% for Fe, and 36.54% for Zn of the total variation under low N conditions, and 12.37% of grain yield, 55.92% for Fe, and 30.46% for Zn under optimum conditions. The response of genotypes to low N conditions was highly variable, confirming the influence of the N levels on Fe and Zn concentration and grain yield performance. In general, the performances of the genotype among the N conditions are quite different; therefore, environment-based variety development for Fe and Zn concentration in grain will be advantageous.

## Figures and Tables

**Figure 1 plants-12-01463-f001:**
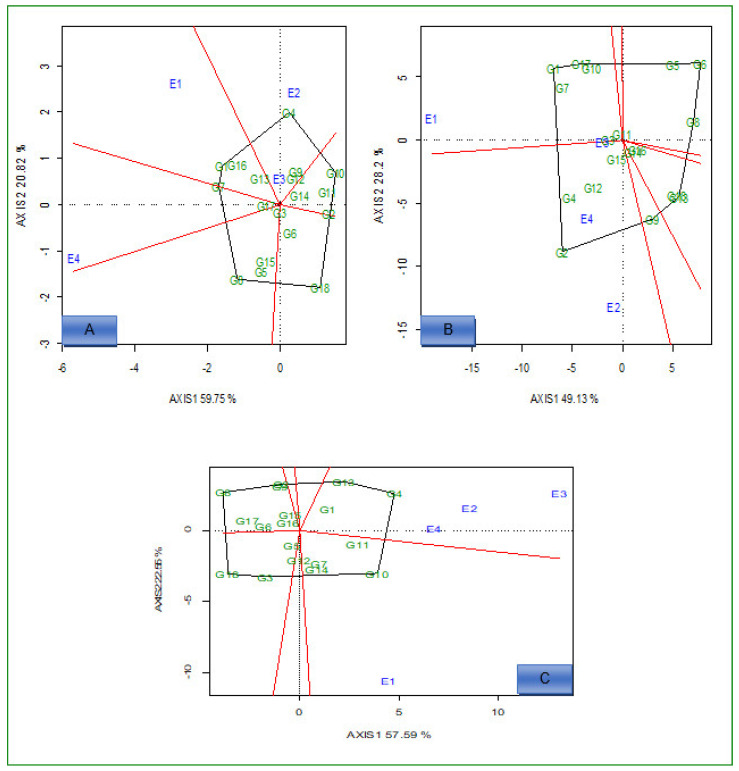
“Which-won-where” view of the GGE biplot for 18 hybrids evaluated for grain yield (**A**), Fe (**B**), and Zn (**C**) concentration under low N conditions across environments. E1 = Cedara 2017; E2 = Vaalharts 2017; E3 = Cedara 2018; E4 = Potchefstroom 2018.

**Figure 2 plants-12-01463-f002:**
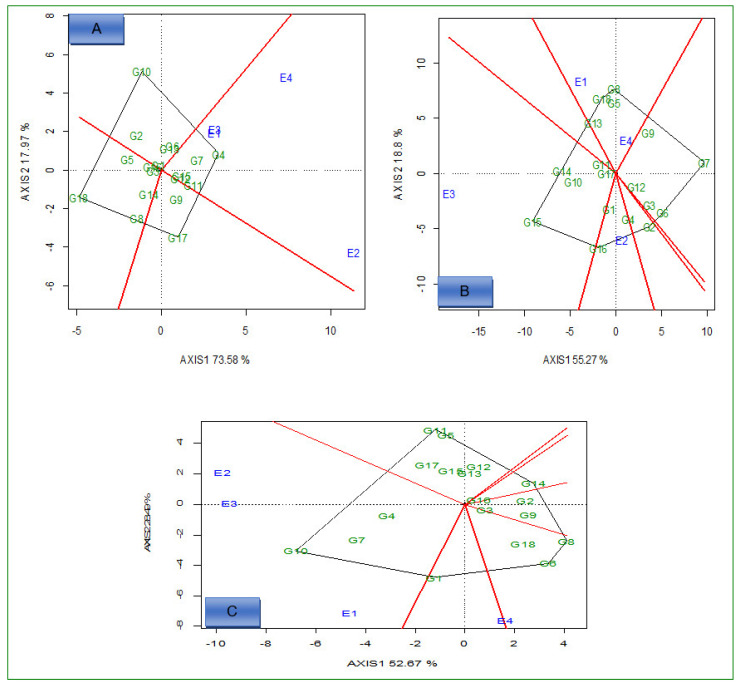
“Which-won-where” view of the GGE biplot for 18 hybrids evaluated for grain yield (**A**), iron (**B**), and zinc (**C**) under optimum conditions across environments. E1 = Cedara, 2017; E2 = Potchefstroom, 2017; E3 = Cedara, 2018; E4 = Potchefstroom, 2018.

**Figure 3 plants-12-01463-f003:**
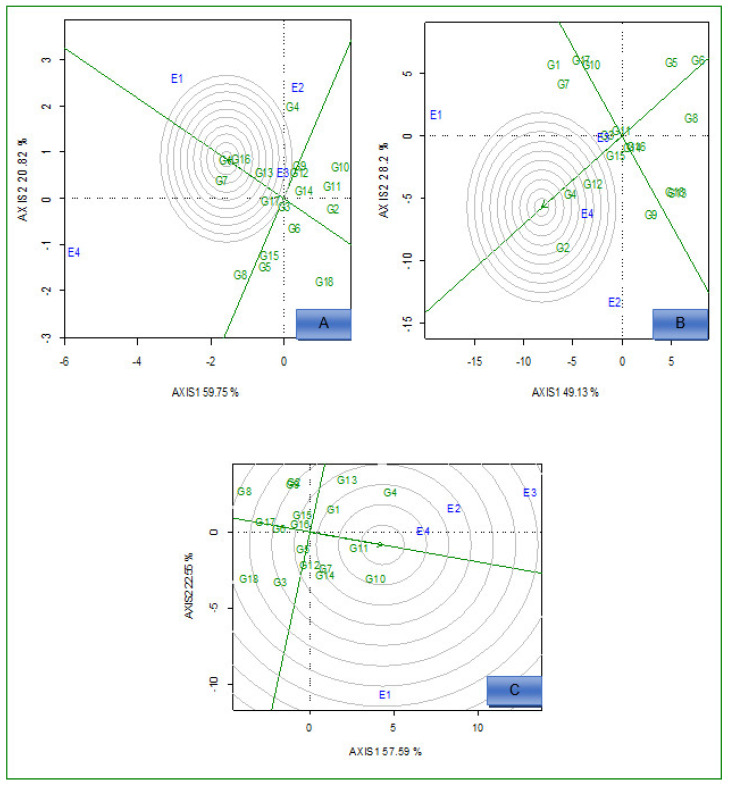
Comparison view of 18 hybrids with the ideal genotype based on average grain yield (**A**), iron (**B**), and zinc (**C**) under low N conditions across environments. E1 = Cedara, 2017; E2 = Cedara, 2017; E3 = Cedara, 2018; E4 = Potchefstroom, 2018.

**Figure 4 plants-12-01463-f004:**
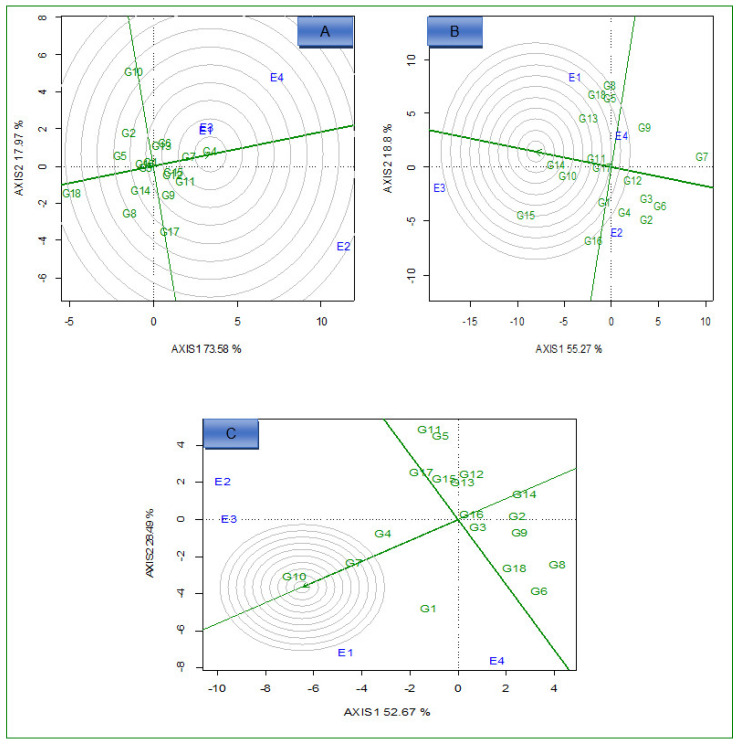
Comparison view of 18 hybrids with the ideal genotype based on average grain yield (**A**), iron (**B**), and zinc (**C**) under optimum N conditions across environments. E1 = Cedara, 2017; E2 = Cedara, 2017; E3 = Cedara, 2018; E4 = Potchefstroom, 2018.

**Figure 5 plants-12-01463-f005:**
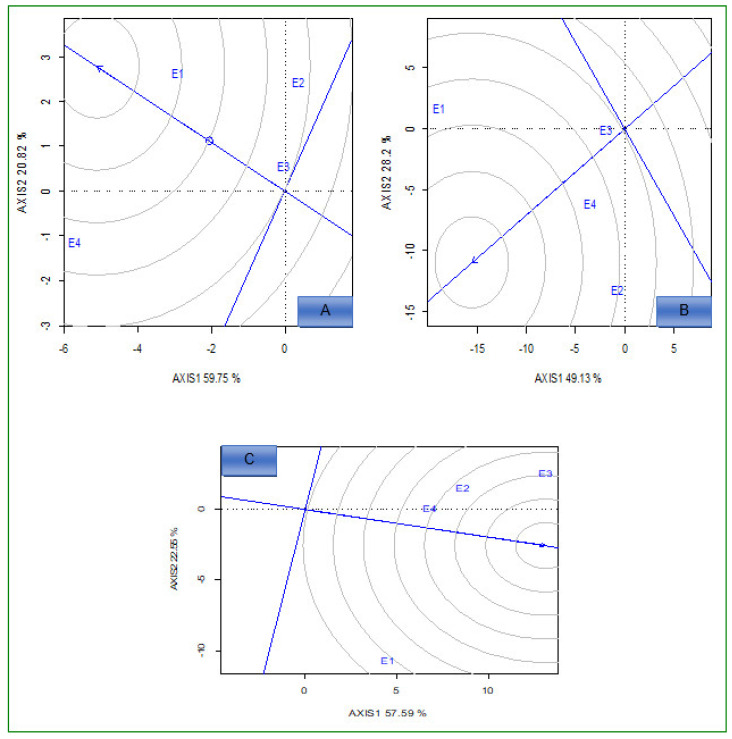
GGE biplot showing the ideal environment for18 hybrids evaluated for grain yield (**A**), iron (**B**), and zinc (**C**) under low N conditions across environments.

**Figure 6 plants-12-01463-f006:**
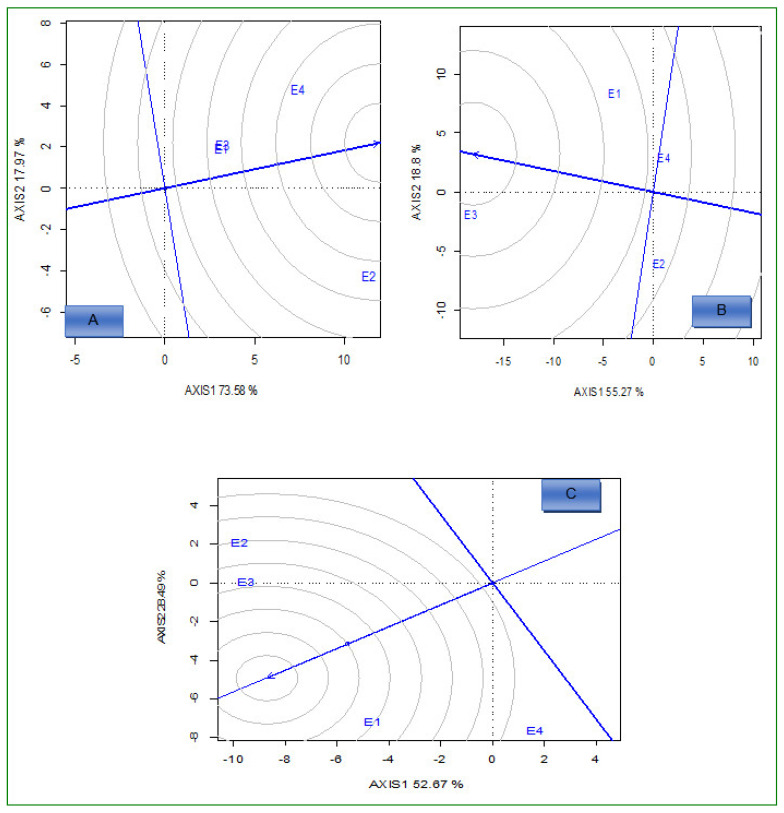
GGE biplot showing the ideal environment for 18 hybrids evaluated for grain yield (**A**), iron (**B**), and zinc (**C**) under optimum N conditions across environments.

**Table 1 plants-12-01463-t001:** AMMI analysis of variance for grain yield, Fe and Zn of 18 maize hybrids evaluated at four trial x treatment combinations (Cedara and Potchefstroom 2017 and 2018) for optimum N conditions.

Grain Yield						Iron				Zinc			
			Sum of Squares Explained (%)				Sum of Squares Explained (%)				Sum of Squares Explained (%)		
Source of Variation	DF	MS	Total VE	GEI E	GEI Cum	MS	Total VE	GEI E	GEI Cum	MS	Total VE	GEI E	GEI Cum
Treatments	71	25.04				22.86				20.61			
Genotypes	17	19.23 **	17.87			19.85 **	20.31			19.67 **	22.03		
Environments	3	408.28 **	66.95			118.82 **	21.45			222.15 **	43.92		
Replication	1	0.29				1.00				0.59			
Interactions	51	4.43 **	12.37			18.22 **	55.92			9.06 **	30.46		
IPCA 1	19	7.93 **		66.78	66.78	28.79 **		58.88	58.88	15.86 **		65.17	65.17
IPCA 2	17	2.97 **		22.32	89.11	13.87 **		25.38	84.26	7.47 **		27.47	92.65
Residuals	15	1.64				9.75				2.27			
Error	68	0.75				0.51				0.77			

** *p* < 0.001, Total VE = total variation explained, GEI E = GEI explained, GEI cum = GEI cumulative, SS = sum of squares, MS = mean squares, DF = degrees of freedom.

**Table 2 plants-12-01463-t002:** AMMI analysis of variance for grain yield, Fe and Zn of 18 maize hybrids evaluated in four trial by treatment combinations (Cedara and Vaalharts 2017 and Cedara and Potchefstroom 2018) for low N conditions.

		Grain Yield				Iron				Zinc			
			Sum of Squares Explained (%)				Sum of Squares Explained (%)				Sum of Squares Explained (%)		
Source of Variation	DF	MS	Total VE	GEI E	GEI Cum	MS	Total VE	GEI E	GEI Cum	MS	Total VE	GEI E	GEI Cum
Treatments	71	2.49				32.83				17.66			
Genotypes	17	2.93 **	24.58			30.42 **	21.87			33.22 **	39.87		
Environments	3	12.91 **	19.77			268.06 **	34.01			57.21 **	12.11		
Replication	1	0.50				0.18				3.79			
Interactions	51	1.76 **	49.71			19.80 **	42.71			10.15 **	36.54		
IPCA 1	19	3.10 **		65.62	65.62	31.41 **		59.09	59.09	14.73 **		54.07	54.07
IPCA 2	17	1.33 **		25.19	90.81	17.84 **		30.02	89.12	11.21 **		36.79	90.87
Residuals	15	0.53				7.33				3.15			
Error	68	0.26				0.48				2.18			

** *p* ≤ 0.001, Total VE = total variation explained, GEI E = GEI explained, GEI cum = GEI cumulative, SS = sum of squares, MS = mean squares, DF = degrees of freedom.

**Table 3 plants-12-01463-t003:** AMMI stability value (ASV), yield stability index (YSI), iron stability index (FSI), zinc stability index (ZSI), its ranks, and IPCA under low N conditions across environments.

	Grain Yield							Fe Concentration in Grain							Zn Concentration in Grain						
Gen	IPCA1	IPCA2	ASV	rASV	GY	rGY	YSI	IPCA1	IPCA2	ASVF	rASV	Fe	rFe	FSI	IPCA1	IPCA2	ASVZ	rASV	Zn	rZn	ZSI
G1	−0.59	−0.23	1.55	13	3.68	2	15	−1.94	0.18	3.83	18	16.34	8	26	0.68	0.00	1.00	8	19.48	7	15
G2	0.61	0.03	1.60	14	1.81	17	31	−0.11	1.30	1.32	8	19.44	1	9	0.80	−0.30	1.22	12	17.39	13	25
G3	−0.10	−0.17	0.31	2	2.43	12	14	−0.49	1.20	1.54	9	15.40	12	21	−1.29	0.89	2.09	15	17.69	12	27
G4	0.46	−0.87	1.49	12	2.84	5	17	−0.49	1.29	1.61	10	18.22	2	12	1.55	−0.09	2.29	18	22.15	2	20
G5	−0.54	0.49	1.49	11	2.52	9	20	0.12	−0.25	0.35	1	12.60	17	18	−0.37	−1.07	1.20	11	18.34	9	20
G6	0.21	0.75	0.93	7	2.61	6	13	0.61	−0.54	1.31	7	11.80	18	25	−0.24	0.81	0.88	5	17.07	15	20
G7	−0.66	0.12	1.73	15	3.74	1	16	−1.23	−1.63	2.92	15	17.93	3	18	−0.56	−1.04	1.33	13	19.72	6	19
G8	−1.04	0.15	2.70	17	2.61	7	24	1.11	−0.47	2.23	11	13.56	16	27	0.14	−0.77	0.80	4	15.11	18	22
G9	0.22	−0.66	0.88	6	2.32	14	20	1.31	0.16	2.58	13	16.71	6	19	0.77	−0.29	1.17	10	17.32	14	24
G10	1.16	0.54	3.08	18	2.40	13	31	−1.03	−1.43	2.48	12	16.25	9	21	−0.30	0.19	0.48	3	22.39	1	4
G11	0.75	0.02	1.95	16	2.04	16	32	0.06	−0.40	0.41	2	16.43	7	9	0.14	0.96	0.98	7	21.34	3	10
G12	0.32	−0.22	0.87	5	2.51	10	15	−0.08	0.66	0.68	4	17.78	4	8	−0.68	0.52	1.12	9	19.03	8	17
G13	0.00	0.26	0.26	1	3.30	4	5	1.49	0.02	2.93	16	14.96	15	31	1.41	0.44	2.12	16	19.89	5	21
G14	0.22	−0.35	0.66	3	2.20	15	18	0.21	0.51	0.66	3	15.52	11	14	−0.72	−1.70	2.00	14	19.92	4	18
G15	−0.46	0.34	1.25	10	2.48	11	21	0.26	−0.93	1.06	6	17.20	5	11	0.28	0.88	0.97	6	17.98	10	16
G16	−0.43	−0.37	1.17	9	3.40	3	12	0.40	−0.31	0.84	5	15.81	10	15	0.12	−0.06	0.19	1	17.86	11	12
G17	−0.32	−0.38	0.93	8	2.58	8	16	−1.61	0.57	3.23	17	15.00	14	31	−0.23	0.19	0.39	2	16.38	16	18
G18	0.18	0.56	0.73	4	1.55	18	22	1.42	0.08	2.80	14	15.01	13	27	−1.52	0.46	2.28	17	16.13	17	34

**Table 4 plants-12-01463-t004:** AMMI stability value (ASV), yield stability index (YSI), iron stability index (FSI), zinc stability index (ZSI), its ranks, and IPCA under optimum conditions across environments.

	Grain Yield							Fe Concentration in Grain							Zn Concentration in Grain						
Gen	IPCA1	IPCA2	ASV	rASV	GY	rGY	YSI	IPCA1	IPCA2	ASVF	rFSV	Fe	rFe	FSI	IPCA1	IPCA2	ASVZ	rZSV	Zn	rZn	ZSI
G1	0.17	0.07	0.52	4	6.51	11	15	0.72	−1.02	1.95	13	21.36	1	14	−0.79	0.66	1.98	11	22.52	3	14
G2	0.78	−0.43	2.38	14	5.45	15	29	0.72	−0.89	1.89	11	17.10	13	24	−0.53	0.64	1.42	7	19.23	15	22
G3	0.09	0.01	0.27	1	6.07	12	13	0.87	−0.61	2.11	14	17.57	12	26	−0.27	0.23	0.68	1	20.08	6	7
G4	−0.79	−0.88	2.52	15	9.33	1	16	−0.05	−0.63	0.64	2	16.91	15	17	0.57	−0.59	1.49	8	21.92	4	12
G5	0.59	−0.16	1.78	10	4.79	17	27	0.20	1.10	1.20	6	19.11	7	13	1.11	0.08	2.64	15	19.31	14	29
G6	0.1	−0.37	0.48	3	7.18	6	9	0.96	−0.61	2.31	15	16.53	16	31	−1.59	0.54	3.82	18	20.01	8	26
G7	−0.55	−0.82	1.83	12	8.2	2	14	2.20	0.08	5.11	18	15.92	18	36	0.51	−0.42	1.28	6	22.99	2	8
G8	−0.07	1.41	1.43	9	5.2	16	25	0.05	1.43	1.43	7	18.66	8	15	−1.40	0.10	3.33	17	19.05	17	34
G9	−0.59	0.32	1.80	11	7.09	7	18	0.67	0.73	1.72	9	16.96	14	23	−0.66	−0.38	1.61	9	19.12	16	25
G10	1.75	−0.3	5.24	18	6.53	10	28	−0.83	−0.21	1.95	12	20.14	3	15	0.93	−0.80	2.34	13	24.34	1	14
G11	−0.61	0.27	1.83	13	8.16	3	16	−0.71	0.34	1.68	8	17.60	11	19	1.11	1.68	3.13	16	20.02	7	23
G12	−0.31	0.39	1.00	7	7.58	4	11	0.03	0.00	0.06	1	16.33	17	18	0.34	0.52	0.96	4	19.55	13	17
G13	0.14	−0.46	0.63	6	7.09	8	14	−0.26	0.76	0.98	4	19.80	4	8	0.38	0.19	0.91	3	19.66	10	13
G14	−0.18	0.03	0.54	5	5.67	14	19	−1.51	0.20	3.50	16	19.25	6	22	−0.29	−0.28	0.74	2	18.50	18	20
G15	−0.37	0.15	1.11	8	7.55	5	13	−2.13	−0.71	4.98	17	20.44	2	19	0.64	−0.60	1.62	10	19.82	9	19
G16	0.13	−0.09	0.39	2	6.07	13	15	−0.53	−1.21	1.73	10	18.60	9	19	0.05	−1.01	1.02	5	19.63	11	16
G17	−1.29	0.14	3.87	17	6.62	9	26	−0.38	0.10	0.89	3	18.12	10	13	0.86	0.30	2.07	12	20.33	5	17
G18	1.00	0.73	3.07	16	2.38	18	34	−0.02	1.15	1.16	5	19.72	5	10	−0.97	−0.86	2.45	14	19.62	12	26

**Table 5 plants-12-01463-t005:** List of hybrids used in this study.

Genotypes	Code	Fe and Zn of Parents	Breeding Status of the Genetic Materials
CBY075 LM-1574 × CBY358 LM-1857	G1	High × high	Hybrid (H1)
CBY075 LM-1574 × CBY104 LM-1603	G2	High × intermediate	Hybrid (H1)
CBY075 LM-1574 × CBY013 LM-1512	G3	High × low	Hybrid (H1)
CBY101 LM-1600 × CBY358 LM-1857	G4	High × high	Hybrid (H1)
CBY101 LM-1600 × CBY104 LM-1603	G5	High × intermediate	Hybrid (H1)
CBY101 LM-1600 × CBY013 LM-1512	G6	High × low	Hybrid (H1)
CBY102 LM-1601 × CBY358 LM-1857	G7	Intermediate × high	Hybrid (H1)
CBY102 LM-1601 × CBY104 LM-1603	G8	Intermediate × intermediate	Hybrid (H1)
CBY102 LM-1601 × CBY013 LM-1512	G9	Intermediate × low	Hybrid (H1)
CBY359 LM-1858 × CBY358 LM-1857	G10	Intermediate × high	Hybrid (H1)
CBY359 LM-1858 × CBY104 LM-1603	G11	Intermediate × intermediate	Hybrid (H1)
CBY359 LM-1858 × CBY013 LM-1512	G12	Intermediate × low	Hybrid (H1)
CBY017 LM-1516 × CBY358 LM-1857	G13	Low × high	Hybrid (H1)
CBY017 LM-1516 × CBY104 LM-1603	G14	Low × intermediate	Hybrid (H1)
CBY017 LM-1516 × CBY013 LM-1512	G15	Low × low	Hybrid (H1)
CBY014 LM-1513 × CBY358 LM-1857	G16	Low × high	Hybrid (H1)
CBY014 LM-1513 × CBY104 LM-1603	G17	Low × intermediate	Hybrid (H1)
CBY014 LM-1513 × CBY013 LM-1512	G18	Low × low	Hybrid (H1)

**Table 6 plants-12-01463-t006:** Soil analysis of experimental sites used.

Minerals	Soil Depth	Potchefstroom		Cedara		Vaalharts	
		Optimal	Low N	Optimal	Low N	Optimal	Low N
		2016–2017	2016–2017	2016–2017	2016–2017	2016–2017	2016–2017
Fe (mg kg^−1^)	30 cm	11.9	10.0	13.5	9.6	7.0	5.9
	60 cm	10.6	8.4	11.9	10.1	6.6	5.9
Zn (mg kg^−1^)	30 cm	9.4	9.0	1.3	3.2	3.3	2.5
	60 cm	8.6	5.6	1.4	2.2	2.9	2.3
P (mg kg^−1^)	30 cm	27.9	15.5	11.7	12.8	52.3	32.4
	60 cm	35.7	12.6	10.5	10.1	44.7	29.3
K (mg kg^−1^)	30 cm	278.5	198.4	77.0	174.5	123	163
	60 cm	314.9	209.7	70.5	120.0	114	149
Ca (mg kg^−1^)	30 cm	830.0	666.0	513.0	699.0	436	535
	60 cm	952.0	887.0	511.0	694.0	402	500
Mg (mg kg^−1^)	30 cm	384.9	328.5	99.0	166.0	141	174
	60 cm	440.7	438.9	99.5	154.0	128	169
Mn (mg kg^−1^)	30 cm	38.9	35.1	3.6	3.4	11.1	13.2
	60 cm	43.8	26.9	3.6	2.3	9.2	13.1
Soil pH	30 cm	6.5	6.1	4.3	4.4	6.0	6.3
	60 cm	6.6	6.0	4.4	4.5	6.1	6.3

## Data Availability

Data are available from the authors.

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
