# Peer review of "Genotype by Environment Interaction in Grain Iron and Zinc Concentration and Yield of Maize Hybrids under Low Nitrogen and Optimal Conditions"

_plants, 2023, doi:10.3390/plants12071463_

Round 1

Reviewer 1 Report

The experiment is well performed and the theme is novel and very interesting, but minor corrections are needed.

- Line 16: "concentration" or "content"? please see lines 17, 20, 22, 23, and so on.

- Line 19: environmental?

- Lines 23-24: please write the full names of the abbreviations "AMMI, FSI, and ZSI" for the first time.

- Line 38: superscript the "-1" on g.

- Lines 119-120: please write some details for the standard agronomic practices applied.

Author Response

Dear reviewer,

Thank you so much for your significant contribution. As per your comments, all issues are addressed.

Thanks, 

Mekonnen

Reviewer 2 Report

I have enjoyed editing this article which shed light on an important aspect of research (biofortification in Maize).  

There few needed corrections and clarifications:

- Please revise the sentences which contain Fe and Zn contents of concentrations  (not Fe and Zn content and concentration in singular). 

- It will be good to clarify if the grain yield is estimated based on plot basis or on the five sampled plant per entry? if using the later, then you need to justify the appropriateness of this approach.

- Other corrections needed are included in the edited manuscript

Author Response

Dear reviewer,

 Thank you very much for your valuable contribution to our manuscript. As per your helpful comments, all comments are accepted and have been improved.

Regards,

Mekonnen
